# Synthesis of Second-Generation Analogs of Temporin-SHa Peptide Having Broad-Spectrum Antibacterial and Anticancer Effects

**DOI:** 10.3390/antibiotics13080758

**Published:** 2024-08-11

**Authors:** Arif Iftikhar Khan, Shahzad Nazir, Muhammad Nadeem ul Haque, Rukesh Maharjan, Farooq-Ahmad Khan, Hamza Olleik, Elise Courvoisier-Dezord, Marc Maresca, Farzana Shaheen

**Affiliations:** 1Third World Center for Science and Technology, H. E. J. Research Institute of Chemistry, International Center for Chemical and Biological Sciences, University of Karachi, Karachi 75270, Pakistan; khanformanite@gmail.com (A.I.K.); nazirshahzad39@gmail.com (S.N.); nadeem.and.chem@gmail.com (M.N.u.H.); rukeshmaharjan2013@gmail.com (R.M.); farooq.khan@iccs.edu (F.-A.K.); 2Aix Marseille Univ, CNRS, Centrale Med, ISM2, 13013 Marseille, France; hamza.olleik@live.com (H.O.); elise.courvoisier-dezord@univ-amu.fr (E.C.-D.)

**Keywords:** antimicrobial peptide, temporin SHa, anticancer

## Abstract

Antimicrobial peptides (AMPs) are a promising class of therapeutic alternatives with broad-spectrum activity against resistant pathogens. Small AMPs like temporin-SHa (**1**) and its first-generation analog [G10a]-SHa (**2**) possess notable efficacy against Gram-positive and Gram-negative bacteria. In an effort to further improve this antimicrobial activity, second-generation analogs of **1** were synthesised by replacing the natural glycine residue at position-10 of the parent molecule with atypical amino acids, such as D-Phenylalanine, D-Tyrosine and (2-Naphthyl)-D-alanine, to study the effect of hydrophobicity on antimicrobial efficacy. The resultant analogs (**3**–**6**) emerged as broad-spectrum antibacterial agents. Notably, the [G10K]-SHa analog (**4**), having a lysine substitution, demonstrated a 4-fold increase in activity against Gram-negative (*Enterobacter cloacae* DSM 30054) and Gram-positive (*Enterococcus faecalis* DSM 2570) bacteria relative to the parent peptide (**1**). Among all analogs, [G10f]-SHa peptide (**3**), featuring a D-Phe substitution, showed the most potent anticancer activity against lung cancer (A549), skin cancer (MNT-1), prostate cancer (PC-3), pancreatic cancer (MiaPaCa-2) and breast cancer (MCF-7) cells, achieving an IC_50_ value in the range of 3.6–6.8 µM; however, it was also found to be cytotoxic against normal cell lines as compared to [G10K]-SHa (**4**). Peptide **4** also possessed good anticancer activity but was found to be less cytotoxic against normal cell lines as compared to **1** and **3**. These findings underscore the potential of second-generation temporin-SHa analogs, especially analog **4,** as promising leads to develop new broad-spectrum antibacterial and anticancer agents.

## 1. Introduction

The rise of drug-resistant infections is a serious challenge to current antimicrobial treatments. Failures of antibiotic drugs to effectively eliminate the resistant pathogens necessitates an urgency to develop new strategies. Antimicrobial peptides (AMPs) offer a promising solution to this problem as they can circumvent microbial resistance, a phenomenon that has rendered many traditional antibiotic drugs ineffective [1]. These peptides possess membrane lytic properties and specifically target the microorganisms but not the mammalian cells. Recent studies have also uncovered an additional role of natural AMPs as a first line of defence across all multicellular organisms [2]. For example, muramyl dipeptide-based molecules have been designed to activate innate immune systems against microbial infections [3,4]. Many AMPs exhibit broad-spectrum activity towards microorganisms and even cancer cells [5]. Furthermore, peptides have lower toxicity than many conventional antibiotics and are less likely to cause adverse effects [6].

Natural AMPs can be found in multicellular organisms like frog skin, which contains more than 300 AMPs, including temporin [7]. The cationic nature and amphipathicity of temporin peptides makes them highly active against numerous pathogens, including bacteria, viruses, parasites and filamentous fungi [8,9]. So far, this family extends by up to 130 peptides. Among this family, temporin-Sha, isolated from the skin secretions of the European Sahara frog *Pelophylax saharicus*, comprises brief chains of 10 to 13 amino acids and is famous for its broad-spectrum antimicrobial activity. Importantly, this peptide and many of its analogs never manifested any cytotoxicity towards mammalian cell lines, including THP-1 monocytes and HepG2 cell lines [9]. Temporin-SHa has also been used as an antimicrobial coating agent [10] against Gram-positive bacteria like *Listeria ivanovii* [11]. This peptide was also found to be safe and the best candidate against fluconazole-resistant *Candida albicans* biofilm [12].

Our research group had previously synthesised linear analogs of temporin-SHa (**1**) to study their anticancer potential. The [G10a]-SHa analog (**2**) was able to suppress the proliferation of breast cancer cells [13]. Furthermore, we assessed the efficacy of some of these analogs against drug-resistant bacterial strains, especially methicillin-resistant *Staphylococcus aureus* (MRSA) [14] and *Helicobacter pylori* [15]. These two distinct investigations specifically examined the antibacterial efficacy of the analogs against clinically relevant bacterial strains. In another study, the potential of temporin-SHa dendrimers was evaluated in treating both cancer and bacterial infections [16]. Based on our previous results, we decided to synthesise second-generation analogs of temporin-SHa (**1**) via the SPPS method to assess the impact of cationic as well as hydrophobic residues on antimicrobial potency and anticancer activity. In recent years, several novel peptides have been synthesised, such as 45-membered macrocyclic thioether peptide BMS-986189. The yield of this peptide was improved 28-fold with a new purification approach in the SPPS method [17]. Additionally, the quantification of piperidine by UV-Vis monitoring system has proven an inexpensive and easy to implement method, reducing bothwaste volumes and process time [18]. Furthermore, a new antibiotic peptide—teixobatin—was recently found to be highly effective against numerous bacterial pathogens [19]. For this peptide, new catch-and-release purification methods, such as reactive capping purification (RCP) were employed, allowing chromatography-free purification on solid support [20]. We have applied a similar SPPS strategy for the synthesis of linear second-generation analogs of temporin-SHa. All the analogs were fully characterised by CD, NMR, UV, and IR spectroscopy.

## 2. Results

Temporin-SHa (1) and its analogs (**2**–**6**) were synthesised using a solid-phase peptide synthesis (SPPS) approach involving Fmoc chemistry, as reported earlier [13]. The synthetic route for the second-generation analogs (**3**–**6**) of temporin-SHa (**1**) is shown in Figure 1. These second-generation analogs (**3**–**6**) were synthesised on Rink amide resin. Briefly, the resin (0.5 mmol/g) was soaked in DMF, followed by sequential coupling of Fmoc-protected amino acid (5 equiv). For this coupling, ethyl cyano(hydroxyimino) acetate (5 equiv) and N,N-diisopropylcarbodiamide (DIC) (5 equiv) were used. The progress of each coupling step was monitored with a Kaiser test. After synthesis, peptides were cleaved from the resin using a trifluoroacetic acid (TFA) cocktail (trifluoroacetic acid–water–ethane-1, 2-dithiol–triisopropylsilane 88:5:5:2). The crude mixture was concentrated via rotary evaporator and purified by reverse-phase high-performance liquid chromatography (RP-HPLC) using acetonitrile/water as the mobile phase. The purity of each peptide was then established with UPLC. Furthermore, the structure of all the compounds (**1**–**6**) was confirmed using 1D (^1^H/^13^C/DEPT), 2D (COSY/HSQC/HMBC/NOESY) NMR spectroscopy and mass spectrometry. Detailed sequences of each peptide (**1**–**6**) are shown in Table 1. The Appendix A includes a comprehensive SPPS scheme (Appendix A), the UPLC profile of each compound, ESI-MS data, ^1^H-NMR data in tabular form, and the corresponding spectra of the synthesised compounds. These figures Appendix A provide detailed information on the synthesis and characterisation of these peptides.

### 2.1. Amphiphilicity of Peptides

The relative hydrophobicity and quality of the peptides were assessed using UPLC (Figure 1), The retention times of these peptides were analysed.

The net charge of the four compounds at a neutral pH was calculated using a free peptide calculator available on the website of China Peptides Co., Ltd., a company located in Shanghai, China (https://www.chinapeptides.com/; accessed on 15 December 2023). The retention times of the peptides were found to be in the order of 3.489, 3.529, 3.553, 3.806 and 4.749, indicating their hydrophobic order from least to most hydrophobic analog. The key physiochemical parameters of SHa analogs, along with their overall yields, are presented in Table 2.

### 2.2. Secondary Structures Determination

Circular dichroism of temporin SHa, [G10a]-SHa and newly synthesised second-generation analogs of [G10a]-SHa in 20 mM SDS are shown in Figure 2. In lipidic environments mimicked by SDS, peptides adopt an alpha helical structure due to their amphipathic nature. This structural feature is important for their antimicrobial activity, allowing for their insertion into bacterial membranes. The hydrophobic region of the peptide interacts with the hydrophobic portion of the lipid while the hydrophilic region remains exposed to the aqueous environment [21]. The percentage of alpha helix, as well as the percentage of other secondary structures, was determined from CD data using Bestsel software (https://bestsel.elte.hu/index.php, (accessed on 22 June 2024)) (Table 3). Temporin-SHa (**1**) adopted a 49.5% α-helical structure, while its first-generation analog [G10a]-SHa (**2**) acquired a 82.4% alpha helical structure. The 84.6%, 88.7%, 4.7%, and 53.8% alpha helical structures were acquired for second-generation analogs **3**, **4**, **5**, and **6**, respectively.

### 2.3. NMR Spectroscopic Analysis

The Proton NMR spectrum of SHa consists of six distinct regions. The first region ranges from 0.3 ppm to 1.0 ppm, showing methyl protons, except the methionine methyl, which usually appears at 2.0 ppm. The second region ranges from 1.0 ppm to 2.4 ppm, showing methylene protons of all aliphatic amino acids. The third region ranges from 2.7 ppm to 3.3 ppm, showing six methylene protons of aromatic amino acids, whereas the fourth region ranges from 3.6 ppm to 4.6 ppm, showing the alpha protons. The fifth region contains the ten protons of the two aromatic rings in the range of 7.1 ppm to 7.4 ppm. The sixth region exhibits a range of chemical shifts from 7.7 ppm to 8.6 ppm, indicating the presence of amide protons. The Proton NMR spectrum of [G10a]-SHa (**2**) contains a similar number of regions, the first region ranging from 0.3 ppm to 0.9 ppm, which clearly depicts the methyl protons. On the contrary, the methyl protons of methionine appear at 2.0 ppm. Methylene protons of aliphatic amino acids appear from 1.1 ppm to 2.4 ppm, whereas aromatic amino acids display their methylene protons in the regions of 2.7 ppm to 3.3 ppm. Alpha protons of all amino acids appear in the range of 3.6 ppm to 4.6 ppm. In addition, the region of aromatic protons starts at 7.1 ppm and ends at 7.4 ppm, while the amide protons of all amino acids appear at 7.7 ppm to 8.6 ppm. [G10f]-SHa (**3**) and [G10y]-SHa (**6**) show the minor differences in the chemical shifts of protons, apart from in the aromatic region, where 15 protons display their shifts from 7.1 ppm to 7.4 ppm. The rest of the protons of amino acids show nearly similar chemical shifts due to their same position in the peptide sequence, except for the hydroxyl protons of the aromatic portion of tyrosine that appears at 9.17 ppm. The ^1^H NMR spectrum of [G10n]-SHa (**5**) shows 20 distinct aromatic proton peaks that appear within the range of 7.1 ppm to 7.4 ppm. On the other hand, [G10K]-SHa (**4**) shows similar chemical shifts for methyl and methylene protons. Region from 3.6 ppm to 4.6 ppm show alpha protons, while 10 protons of the two aromatic rings display their peaks in the range of 7.1 ppm to 7.4 ppm. The last region ranges from 7.7 ppm to 8.6 ppm, which shows only amide protons.

### 2.4. Antibacterial Activities of the Peptides

The study compared the antibacterial activity of natural peptide temporin-SHa (**1**) and its first-generation analog [G10a]-SHa (**2**), as well as four second-generation analogs, [G10f]-SHa (**3**), [G10K]-SHa (**4**), [G10n]-SHa (**5**), and [G10y]-SHa (**6**), against various Gram-negative and Gram-positive bacterial strains (Table 4 and Table 5). The results showed that **1** was active against all bacterial strains tested, with MIC values ranging from 1.56 to 50 µM. [G10a]-SHa (**2**) was also active against several bacterial strains, but, contrary to the parent peptide, temporin-SHa (**1**), it was not effective against *E. cloacae*, *P. aeruginosa*, and *E. faecium* (MIC > 100 µM).

The activity of the second-generation analogs of temporin-SHa varied depending on the bacterial strain tested. [G10K]-SHa (**4**) was at least four times more active than [G10a]-SHa (**2**) against A. baumannii, *P. aeruginosa*, K. pneumonia, and E. coli, and equally active compared to **2** against H. pylori, B. subtilis, and S. aureus. Moreover, **4** showed at least 16-times more potent activity against *E. faecium*, and was about eight times more active than **2** against *E. cloacae*. On the other hand, [G10y]-SHa (**6**) was found to be equally active as [G10a]-SHa (**2**) against A. baumannii but was less active than **2** against H. pylori, B. subtilis, and S. aureus. Moreover, it showed at least eight times more activity against *E. faecium* and twice the activity compared to **2** when screened against E. faecalis. [G10f]-SHa (**3**) was found to be as equally active as [G10a]-SHa (**2**) against H. pylori, B. subtilis, and S. aureus, less active than **2** against A. baumannii but, 32-times more active against *E. faecium* and eight times more active against E. faecalis compared to [G10a]-SHa (**2**). Finally, the [G10n]-SHa (**5**) derivative was found to be the least active peptide. It mostly had no activity on various strains except B. subtilis, H. pylori, and S. aureus, with MIC values ranging from 50 to 100 µM.

### 2.5. Antiproliferative Activity of the Peptides

Taking into account the previously known anticancer effect of temporin-SHa (**1**) and [G10a]-SHa (**2**) [13], the antiproliferative activity of peptide analogs was then evaluated using both human cancer and normal/non-cancerous cells (Figure 3 and Figure 4 and Table 6 and Table 7).

Temporin-SHa (**1**) was not effective in preventing cancer or normal cell proliferation up to 100 µM (IC_50_ > 100 µM), except for in breast cancer MCF-7 cells. These results are in line with our previous report. [13] On the other hand, the [G10a]-SHa analog (**2**) displayed inhibitory activity against the proliferation of all types of human cancer cells tested, with IC_50_ ranging from 12.1 to 37.2 µM (mean of 18.7 ± 9.0 µM), which confirmed previously published results [13]. Interestingly, when comparing the effect of compound **2** on normal/non cancerous human lung cells versus lung cancerous cells, it was observed that its activity on normal human fibroblast (IMR-90) cells and normal human epithelial lung cells (BEAS-2B) was approximately half as effective as on human lung cancer cells (A549), with IC_50_ values of 33.0 ± 3.1, 22.6 ± 0.7, and 12.1 ± 0.4 µM, respectively. This suggests that [G10a]-SHa (**2**) exhibits relative selectivity towards cancer cells over the normal ones (Table 7). Analog [G10f]-SHa (**3**) showed even greater activity than **2**, with IC_50_ values for cancer proliferation ranging from 3.6 to 19.8 µM. Moreover, this selectivity was also evident when comparing its effects on normal versus cancer lung cells, with IC_50_ of 14.2, 7.8, and 3.6 µM for IMR-90, BEAS-2B, and A549 cells, respectively. The peptide analogs [G10y]-SHa (**6**) and [G10K]-SHa (**4**) were also active on cancer cells with IC_50_ on cancer cells close to the ones found for **2**, ranging from 13.1 to 44.1 µM (mean 26.5 ± 11.3 µM) and from 11.7 to 64.3 µM (mean 35.8 ± 20.2 µM), respectively. Similar to compounds **2** and **3**, we found that **4** and **6** were also more active on cancer cells (A549), with IC_50_ of 32.6 µM for **4** and 74.8 µM for **6**, compared to values grater than 100 µM for normal cells (IMR-90 and BEAS-2B). Notably, as with antibacterial activity, [G10n]-SHa (**5**) was inactive against the tested human cells. Lastly, the assessment of temporin-SHa derivatives on confluent normal and cancer lung cells revealed that, although these analogs effectively inhibited the proliferation of dividing cells, none of them showed toxic effects at least up to a concentration of 100 µM (Figure 5) (CC_50_ > 100 µM).

## 3. Discussion

Temporin-SHa peptides, naturally found in amphibians, have garnered great attention for their antimicrobial efficacy against Gram-positive and Gram-negative bacteria and lower toxicity towards mammalian cells. Their analogs were designed to study the impact of hydrophobicity and unusual amino acid substitutions on chemical and biological properties. Our previous studies of temporin-SHa analogs have already proven the efficacy of D-Ala substituted temporin-SHa analogs against cancer [13] and microbial infections [14,15]. This study has unveiled the promising dual efficacy of second-generation analogs of the parent peptide not only as antimicrobial agents but also potential anticancer therapeutics. In the current study, new analogs were designed by substituting the D-Ala of [G10a]-SHa peptide (**2**) with more hydrophobic amino acids, such as D-Phe, D-Tyr and D-NAL to study the effect of hydrophobicity as well as unusual residues regarding the chemical and biological characteristics of **2**. Furthermore, the effect of positive charge was also witnessed by the insertion of cationic residue in the form of L-Lys at 10th position. The characterisation of peptide analogs was performed by electron spray ionisation mass spectroscopy (ESI), NMR, and circular dichroism (CD) studies.

This study demonstrated that the antibacterial activity of newly synthesised peptide analogs varied depending upon the bacterial strain tested. Specifically, [G10K]-SHa (**4**) was the most promising derivative, showing four-fold increased activity against *A. baumannii*, *P. aeruginosa, K. pneumonia* and *E. coli*, *E. faecium* and *E. faecalis* compared to [G10a]-SHa (**2**). Conversely, the compound [G10y]-SHa (**6**) exhibited enhanced antibacterial efficacy against Gram-positive bacteria *E. faecalis* and *E. faecium* but lower activity against *H. pylori, B. subtilis*, and *S. aureus.* The efficacy of [G10f]-SHa (**3**) was comparable to [G10a]-SHa (**2**) against most strains except for *A. baumannii*; however, a 32-fold increase in activity against *E. faecium* and an eight-fold increase against *E. faecalis* was observed. [G10n]-SHa (**5**) was the least active peptide against most of the bacterial strains, having MIC values in the range of 50–100 µM on *B. subtilis H. pylori*, and *S. aureus*. In terms of anticancer activity, analogs exhibited varying degrees of efficacy across multiple cancer cell lines, with minimal cytotoxicity being observed on normal human cells. Findings of this study were consistent with previously published work [13], wherein temporin-SHa exhibited the activity solely on breast cancer (MCF-7) cells. Conversely, analog **2** exhibited significant inhibitory effects against numerous cancerous cell lines, including breast cancer, liver, ovarian, pancreatic, prostate, skin cancer, and lung cancer cell lines, which corroborated our previous findings [13]. Notably, analog **2** demonstrated relative selectivity on cancer cell lines over normal human fibroblast (IMR-90) cells and normal human epithelial lung cells (BEAS-2B). Furthermore, **3** emerged as the most potent peptide among all analogs, showing higher activity in inhibiting cancer cell proliferation. In addition, the lower IC_50_ values of **3** in cancerous cells as compared to normal cells demonstrated its selectivity against cancer cell lines; however, analogs **4** and **6** demonstrated activity against various cancer cell lines similar to analog **2**. No cytotoxic effect of the analogs was observed on confluent/non-dividing human cells (either normal or cancer ones), demonstrating that the antiproliferative effect of the analogs is due to a specific effect on dividing cells, particularly on cancer ones. Secondary structures seem to play an important role in the biological activities of the analogs. It is known that alpha helical structure is critical for the antibacterial activity of many described AMPs, including temporins. Accordingly, analogs having a percentage of alpha helicity close or superior to native temporin SHa (49.5% of an alpha helical structure for SHa versus 53.8 to 88.7% of an alpha-helical structure for analogs **2**, **3**, **4**, and **6**) display similar or improved antibacterial effects. In contrast, analog **5,** with 4.7% of an alpha helical structure, possesses low to no antibacterial effect. Regarding the antiproliferative effect, the percentage of the alpha helical structure seems again to influence the observed activity. Thus, analog **3,** with 84.6% of alpha-helix is the more active, whereas analog **5,** with the lowest percentage of helicity (4.7%), has low to no effect. The relationship between the percentage of alpha-helix and the antiproliferative effect of analogs is, however, subtler, since analog **4,** with 88.7% of alpha-helix, is less active than analog **3**.

## 4. Materials and Methods

### 4.1. Reagents

The reagents and chemicals used in this study were of high purity, typically ranging between 96 and 98%. Fmoc-protected Rink amide resin with a loading capacity of 0.602 mmol/g and a mesh size of 200–300 was employed. All the Fmoc-protected amino acids and coupling reagents were sourced from Novabiochem^TM^ (Merck, St. Louis, MO, USA), while other chemicals were acquired from Sigma–Aldrich (Merck, St. Louis, MO, USA). HPLC-grade solvents were used throughout the experimental work. Chemical shifts were recorded in ppm. All the synthesised peptides were purified with recycling preparative HPLC (LC-908). For this purpose, SP-120-10 column (C-18) and a solvent system of Acetonitrile/H_2_O (60:40) in 0.08% TFA was employed.

### 4.2. Peptide Synthesis

Temporin-SHa (**1**) and its linear analogs (**2**–**6**) were synthesised via the solid-phase peptide synthesis (SPPS) strategy, as described earlier [13]. The purity of each peptide was established with UPLC. Furthermore, the structure of all the compounds (**1**–**6**) was confirmed using 1D (^1^H/^13^C/DEPT), 2D (COSY/HSQC/HMBC/NOESY) NMR spectroscopy and mass spectrometry. The Appendix A includes a comprehensive SPPS scheme (Appendix A), the UPLC profile of each compound, ESI-MS data, ^1^H-NMR data in tabular form, and the corresponding spectra of the synthesised compounds. These figures Appendix A provide detailed information on the synthesis and characterisation of these peptides.

#### 4.2.1. Synthesis of Temporin-SHa (**1**)

Temporin-SHa (**1**) was synthesised using the above-mentioned SPPS method. Overall yield: 37%; ${\left[\alpha \right]}_{D}^{24}$ = −29 (*c* 0.0007, 60% ACN, 40% H_2_O, 0.082% TFA). UV-Vis. (MeOH) λ_max_ (log ε): 230 (0.982) nm. IR (KBr, cm^−^^1^): 1130.7 (C-O stretching, CN stretching, NH bending), 1520.8 (C-H bending, Aromatic C=C stretching), 1650.1 (NHC=O stretching), 2950.8 (C-H stretching), 3490.0 (OH stretching) and 3280.0 (NH stretching). ^1^H NMR (d_6_-DMSO, 600 MHz): *δ*_H_ 0.76–0.79 (18H, m, (CH_3_)_2_–Leu^2,9,11^), 0.79–0.98 (12H, m, *δ*-CH_3_–IIe^5^, *γ*-CH_3_–IIe^5^, (CH_3_)_2_–Val^6^), 1.24–1.62 (18H, m, CH_2_–Leu^2,9,12^, *γ*-CH–Leu^2,9,12^, *β*-CH –IIe^5^, *γ*-CH_2_–Met^8^, *β*-CH_2_–Lys^11^, *γ*-CH_2_–Lys^11^, *δ*-CH_2_–Lys^11^), 1.69 (1H, m, *γ*-CH,–IIe^5^), 1.90 (1H, m, *β*-CH–Val^6^), 1.99 (3H, s, *δ*-CH_3_–Met^8^), 2.39 (2H, m, *β*-CH_2_–Met^8^), 2.71 (2H, m, Δ-CH_2_–Lys^11^), 2.77–3.10 (4H, dd, CH_2_–Phe^1,13^), 3.55–3.61 (2H, m, CH_2_–Ser^3^), 3.67–3.82 (6H, m, CH_2_–Gly^4,7,10^), 4.04–4.25 (6H, m, *α*-CH–IIe^5^, CH–Lys^11^, *α*-CH–Val^6^, *α*-CH–Leu^9,12^, *α*-CH–Phe^1^), 4.26–4.46 (4H, m, CH–Met^8^, *α*-CH–Phe^13^, *α*-CH–Leu^2^, CH–Ser^3^), 5.06, (1H, t, OH–Ser^3^), 7.22–7.25 (10H, m, CH_Ar_–Phe^1,13^), 7.64–7.98 (5H, m, NH–Met^8^, NH–IIe^5^, NH–Val^6^, NH–Leu^12^, NH–Phe^13^), 8.06–8.11 (5H, m, NH–Lys^11^, NH–Leu^9^, NH–Gly^4,7,10^), 8.18–8.61 (2H, m, NH–Ser^3^, NH–Leu^2^). ^13^C-NMR (d_6_-DMSO, 150 MHz): δ ppm 10.9, 14.6, 15.3, 18.3, 18.5, 19.1, 21.5, 21.6, 22.1, 22.5, 23.0, 23.2, 24.0, 24.1, 24.2, 24.7, 26.6, 28.6, 29.4, 30.3, 31.6, 32.0, 36.5, 37.0, 37.6, 38.7, 40.3, 40.5, 40.6, 41.0, 42.0, 50.9, 51.0, 51.2, 51.9, 53.2, 53.4, 55.1, 55.6, 56.7, 57.1, 58.0, 61.8, 126.2, 127.1, 128.0, 128.5, 129.2, 129.5, 134.8, 137.6, 157.4, 157.6, 157.9, 158.1, 167.7, 168.5, 168.6, 168.8, 170.2, 171.0, 171.1, 171.4, 171.5, 171.9, 172.3, 172.6. HR-MALDI-MS [M + Na]^+^: *m*/*z* calculated for [C_67_H_109_N_15_O_14_S + Na]^+^: 1402.7891; found: 1402.7891.

#### 4.2.2. Synthesis of [G10a]-SHa Analog (**2**)

Temporin-SHa analog [G10a]-SHa (**2**) was synthesised using the above-mentioned Fmoc-SPPS method. Overall yield: 81.2%; ${\left[\alpha \right]}_{D}^{25}$ = −59.53 (*c* 0.001, MeOH). UV-Vis (MeOH) λ_max_ (log ε): 231.00 (2.91) nm. IR (KBr, cm^−^^1^): 1135.78 (C-O stretching, CN stretching, NH bending), 1456.86 (C-H bending, Aromatic C=C stretching), 1654.06 (NHC=O stretching), 2961.78 (C-H stretching), 3445.01 (OH stretching) and 3673.83 (NH stretching). ^1^H NMR (d_6_-DMSO, 600 MHz): *δ*_H_ 0.76–0.79 (18H, m, (CH_3_)_2_–Leu^2,9,12^), 0.79–0.98 (12H, m, *δ*-CH_3_–IIe^5^, *γ*-CH_3_–IIe^5^, (CH_3_)_2_–Val^6^), 1.18 (3H, d, CH_3_–Ala^10^), 1.24–1.62 (18H, m, CH_2_–Leu^2,9,12^, γ-CH–Leu^2,9,12^, *β*-CH –IIe^5^, *γ*-CH_2_–Met^8^, *β*-CH_2_–Lys^11^, *γ*-CH_2_–Lys^11^, *δ*-CH_2_–Lys^11^), 1.69 (1H, m, *γ*-CH–IIe^5^), 1.90 (1H, m, *β*-CH–Val^6^), 1.99 (3H, s, *δ*-CH_3_–Met^8^), 2.39 (2H, m, *β*-CH_2_–Met^8^), 2.71 (2H, m, Δ-CH_2_–Lys^11^), 2.77–3.10 (4 H, dd, CH_2_–Phe^1,13^), 3.54–3.60 (2H, m, CH_2_–Ser^3^), 3.67–3.82 (4H, m, CH_2_–Gly^4,7^), 4.04–4.25 (6H, m, *α*-CH–Phe^1^, *α*-CH–IIe^5^, *α*-CH–Val^6^, *α*-CH–Leu^9,12^, CH–Ala^10^), 4.26–4.46 (5H, m, *α*-CH–Leu^2^, CH–Ser^3^, CH–Met^8^, CH–Lys^11^, CH–Phe^13^), 5.06, (1H, t, OH–Ser^3^), 7.22–7.25 (10H, m, CH_Ar_–Phe^1,13^), 7.64–7.98 (5H, m, NH–Val^6^, NH–Met^8^, NH–Ala^10^, NH–Leu^12^, NH–Phe^13^), 8.06–8.11 (4H, m, NH–Gly^4,7^, NH–Leu^9^, NH–Lys^11^), 8.18–8.60 (3H, d, NH–Leu^2^, NH–Ser^3^, NH–IIe^5^). ^13^C-NMR (d_6_-DMSO, 150 MHz): δ ppm 10.9, 14.6, 15.3, 18.4, 19.2, 21.5, 21.6, 22.1, 22.9, 23.0, 23.2, 24.0, 24.1, 24.2, 26.6, 29.4, 30.2, 31.3, 31.8, 36.5, 37.0, 37.6, 38.8, 39.3, 39.5, 39.6, 39.7, 39.9, 40.0, 40.4, 40.6, 41.0, 41.9, 42.1, 48.3, 50.9, 51.3, 51.4, 52.0, 53.2, 53.5, 55.1, 56.7, 58.0, 61.8, 126.2, 127.1, 128.0, 128.1, 128.5, 129.1, 129.2, 129.5, 134.7, 137.6, 157.7, 157.9, 167.7, 168.5, 168.9, 170.2, 171.0, 171.1, 171.2, 171.4, 171.6, 172.1, 172.7. LRMS (ESI) *m*/*z*: 1394.9 [M + H]^+^.

#### 4.2.3. Synthesis of [G10f]-SHa Analog (**3**)

Second-generation analog [G10f]-SHa (**3**) of temporin-SHa was synthesised using the above-mentioned Fmoc-SPPS method. Overall yield: 76.7%; ${\left[\alpha \right]}_{D}^{25}$ = −47.69 (*c* 0.001, MeOH). UV-Vis. (MeOH) λ_max_ (log ε): 229.00 (3.69) nm, IR (KBr, cm^−^^1^): 1196.34 (C-O stretching, CN stretching, NH bending), 1441.05 (C-H bending, Aromatic C=C stretching), 1660.51 (NHC=O stretching), 2956.31 (C-H stretching), 3292.91 (OH stretching) and 3783.25 (NH stretching). ^1^H NMR (d_6_-DMSO, 600 MHz): *δ*_H_ 0.76–0.79 (18 H, m, (CH_3_)_2_–Leu^2,9,12^), 0.79–0.98 (12H, m, *γ*-CH_3_–IIe^5^, *δ*-CH_3_–IIe^5^, (CH_3_)_2_–Val^6^), 1.24–1.62 (18H, m, CH_2_–Leu^2,9,12^, *γ*-CH–Leu^2,9,12^, *β*-CH –IIe^5^, *γ*-CH_2_–Met^8^, *β*-CH_2_–Lys^11^, *γ*-CH_2_–Lys^11^, *δ*-CH_2_–Lys^11^), 1.69 (1H, m, *γ*-CH–IIe^5^), 1.90 (1H, m, *β*-CH–Val^6^), 1.99 (3H, s, *δ*-CH_3_–Met^8^), 2.39 (2H, m, *β*-CH_2_–Met^8^), 2.71 (2H, m, Δ-CH_2_–Lys^11^), 2.77–3.10 (6H, dd, CH_2_–Phe^1,10,13^), 3.54–3.60 (2H, m, CH_2_–Ser^3^), 3.67–3.82 (4H, m, CH_2_–Gly^4,7^), 4.04–4.25 (5H, m, *α*-CH–Phe^1^*, α*-CH–IIe^5^, *α*-CH–Val^6^, *α*-CH–Leu^9,12^), 4.26–4.46 (6H, m*, α*-CH–Leu^2^, CH–Ser^3^, CH–Met^8^, CH-Phe^10,13^, CH–Lys^11^), 5.05, (1 H, bs, OH–Ser^3^), 7.22–7.25 (15H, m, CH_Ar_–Phe^1,10,13^), 7.64–8.01 (5H, m, NH–Val^6^, NH–Met^8^, NH–Leu^9,12^, NH–Phe^10^), 8.06–8.11 (4H, NH–Gly^4,7^, NH–IIe^5^, NH–Lys^11^), 8.17–8.60 (3H, d, NH–Leu^2^, NH–Ser^3^, NH–Phe^13^). ^13^C-NMR (d_6_-DMSO, 150 MHz): δ ppm 10.9, 14.6, 15.3, 18.3, 19.1, 21.5, 21.6, 22.1, 22.8, 22.9, 23.1, 23.9, 24.0, 24.2, 24.4, 26.6, 29.4, 30.2, 31.3, 31.6, 31.9, 36.5, 36.9, 37.5, 37.8, 37.9, 38.7, 40.6, 40.9, 41.9, 42.1, 51.0, 51.2, 51.3, 51.8, 51.9, 53.1, 53.4, 54.1, 55.1, 56.7, 58.0, 58.8, 61.8, 125.9, 126.2, 127.1, 127.9, 128.5, 129.1, 129.2, 129.5, 134.7, 136.6, 137.6137.7, 157.7, 157.9, 167.7, 168.5, 168.8, 170.2, 170.8, 170.9, 171.0, 171.1, 171.2, 171.4, 171.6, 171.7, 172.7. LRMS (ESI) *m*/*z*: 1470.86/744.1 [M + H + NH_4_]^+2^.

#### 4.2.4. Synthesis of [G10K]-SHa Analog (**4**)

Analog [G10K]-SHa (**4**) of temporin-SHa peptide was synthesised using the above-mentioned Fmoc-SPPS method. Overall yield: 72%. ${\left[\alpha \right]}_{D}^{25}$ = −59.53 (*c* 0.001, MeOH). UV-Vis (MeOH) λ_max_ (log ε): 230.00 (3.67) nm, IR (KBr, cm^−^^1^): 1195.54 (C-O stretching, CN stretching, NH bending), 1441.84 (C-H bending, Aromatic C=C stretching), 1661.79 (NHC=O stretching), 2956.85 (C-H stretching), 3306.46 (OH stretching) and 3786.8 (NH stretching). ^1^H NMR (d_6_-DMSO, 600 MHz): *δ*_H_ 0.76–0.79 (18H, m, (CH_3_)_2_–Leu^2,9,12^), 0.79–0.98 (12H, m, *γ*-CH_3_–IIe^5^, *δ*-CH_3_–IIe^5^, (CH_3_)_2_–Val^6^), 1.24–1.62 (24H, m, CH_2_–Leu^2,9,12^, γ-CH–Leu^2,9,12^, *β*-CH –IIe^5^, *γ*-CH_2_–Met^8^, *β*-CH_2_–Lys^10,11^, *γ*-CH_2_–Lys^10,11^, *δ*-CH_2_–Lys^10,11^), 1.69 (1 H, m, *γ*-CH–IIe^5^), 1.90 (1H, m, *β*-CH–Val^6^), 1.99 (3H, s, *δ*-CH_3_–Met^8^), 2.39 (2H, m, *β*-CH_2_–Met^8^), 2.71 (4H, m, Δ-CH_2_–Lys^10,11^), 2.80–3.10 (4H, dd, CH_2_–Phe^1,13^), 3.55–3.61 (2H, m, CH_2_–Ser^3^), 3.67–3.82 (4H, m, CH_2_–Gly^4,7^), 4.04–4.25 (5H, m, *α*-CH–IIe^5^, *α*-CH–Val^6^, *α*-CH–Leu^9^, CH–Lys^11^, *α*-CH–Leu^12^), 4.26–4.46 (6H, m, *α*-CH–Phe^1^*, α*-CH–Leu^2^, CH–Ser^3^, CH–Met^8^, CH–Lys^10^, CH–Phe^13^), 5.08, (1 H, bs, OH–Ser^3^), 7.10–7.25 (10H, m, CH_Ar_–Phe^1,13^), 7.64–8.01 (5H, m, NH–Val^6^, NH–Met^8^, NH–Lys^10,11^, NH–Leu^12^), 8.06–8.11 (4H, m, NH–Gly^4,7^, NH–IIe^5^, NH–Leu^9^), 8.14–8.60 (3H, m, NH–Leu^2^, NH–Ser^3^, NH–Phe^13^). ^13^C-NMR (d_6_-DMSO, 150 MHz): δ ppm 10.9, 14.6, 15.3, 18.4, 19.1, 21.4, 21.6, 22.2, 22.3, 23.1, 23.2, 24.0, 24.0, 24.1, 24.3, 26.6, 26.7, 29.3, 30.1, 31.1, 36.9, 37.6, 40.6, 40.9, 42.0, 42.1, 51.0, 51.3, 51.4, 51.9, 52.4, 53.2, 53.3, 53.5, 55.2, 56.9, 58.3, 61.8, 126.2, 126.4, 127.1, 128.0, 128.1, 128.5, 129.1, 129.2, 129.5, 134.7, 137.4, 137.6, 157.7, 157.9, 158.1, 158.3, 167.7, 168.6, 168.7, 168.9, 170.3, 171.0, 171.1, 171.2, 171.3, 171.4, 171.5, 171.6, 171.9, 172.1, 172.7, 172.8. LRMS (ESI) *m*/*z*: 1453.1 [M + H]^+^.

#### 4.2.5. Synthesis of [G10n]-SHa Analog (**5**)

Second-generation analog [G10n]-SHa of temporin-SHa peptide was synthesised using the above-mentioned Fmoc-SPPS method. Overall yield: 24%. ${\left[\alpha \right]}_{D}^{25}$ = −115.29 (*c* 0.001, MeOH), UV-Vis. (MeOH) λ_max_ (log ε): 228.00 (3.75) nm. IR (KBr, cm^−^^1^): 1203.43 (C-O stretching, CN stretching, NH bending), 1404.01 (C-H bending, Aromatic C=C stretching), 1685.32 (NHC=O stretching), 3135.02 (C-H stretching), 3406.43 (OH stretching) and 38.03.49 (NH stretching). ^1^H NMR (d_6_-DMSO, 600 MHz): *δ*_H_ 0.76–0.79 (18H, m, (CH_3_)_2_–Leu^2,9,12^), 0.79–0.98 (12H, m, *γ*-CH_3_–IIe^5^, *δ*-CH_3_–IIe^5^, (CH_3_)_2_–Val^6^), 1.24–1.62 (18H, m, CH_2_–Leu^2,9,12^, *γ*-CH–Leu^2,9,12^, *β*-CH –IIe^5^, *γ*-CH_2_–Met^8^, *β*-CH_2_–Lys^11^, *γ*-CH_2_–Lys^11^, *δ*-CH_2_–Lys^11^), 1.69 (1H, m, *γ*-CH–IIe^5^), 1.90 (1H, m, *β*-CH–Val^6^), 1.99 (3H, s, *δ*-CH_3_–Met^8^), 2.39 (2H, m, *β*-CH_2_–Met^8^) 2.71 (2H, m, Δ-CH_2_–Lys^11^), 2.77–3.11 (6H, m, CH_2_–Phe^1,13^*, β*-CH_2_–NAL^10^), 3.55–3.61 (2H, m, CH_2_–Ser^3^), 3.67–3.82 (4H, m, CH_2_–Gly^4,7^), 4.04–4.25 (5H, m*, α*-CH–Phe^1^*, α*-CH–IIe^5^, *α*-CH–Val^6^, *α*-CH–Leu^9,12^), 4.26–4.46 (5H, m, *α*-CH–Leu^2^, CH–Ser^3^, CH–Met^8^, CH–Lys^11^, CH–Phe^13^), 4.64 (1H, m, *α*-CH–Tyr^10^), 5.08, (1H, t, OH–Ser^3^), 7.10–7.25 (15H, m, CH_Ar_–Phe^1,13^, CH_Ar_–NAL^10^), 7.64–8.01 (5H, m, NH–Val^6^, NH–Met^8^, NH-Leu^9,12^, NH–NAL^10^), 8.06–8.11 (4H, m, NH–Gly^4,7^, NH–IIe^5^, NH–Lys^11^), 8.17–8.60 (3H, m, NH–Leu^2^, NH–Ser^3^, NH–Phe^13^). ^13^C-NMR (d_6_-DMSO, 150 MHz): δ ppm 11.0, 14.8, 15.4, 18.5, 19.3, 21.6, 21.8, 22.2, 22.6, 22.7, 23.1, 23.3, 23.9, 24.2, 24.4, 26.7, 29.6, 30.4, 37.7, 37.9, 43.8, 49.3, 51.2, 51.5, 51.6, 52.3, 53.4, 53.7, 54.2, 54.5, 55.6, 57.0, 58.4, 60.5, 61.9, 63.4, 121.8, 123.8, 125.6, 126.1, 126.5, 126.6, 127.0, 127.4, 127.6, 127.9, 128.2, 128.3, 128.8, 129.4, 129.7, 130.6, 134.8, 135.5, 136.1, 137.7, 138.0, 138.9, 142.6, 145.4, 154.8, 158.2, 158.4, 159.3, 168.3, 168.9, 169.1, 170.5, 171.1, 171.2, 171.3, 171.5, 171.8, 172.0, 172.1, 173.1, 181.0. LRMS (ESI) *m*/*z*: 1521.2 [M + H]^+^.

#### 4.2.6. Synthesis of [G10y]-SHa Analog (6)

Second-generation analog [G10y]-SHa (**6**) of temporin-SHa peptide was synthesised using the above-mentioned Fmoc-SPPS method. Overall yield: 79%; ${\left[\alpha \right]}_{D}^{25}$ = −77.83 (*c* 0.001, MeOH), UV-Vis. (MeOH) λ_max_ (log ε): 230.00 (3.80) nm. IR (KBr, cm^−^^1^): 1196.26 (C-O stretching, CN stretching, NH bending), 1441.52 (C-H bending, Aromatic C=C stretching), 1641.51 (NHC=O stretching), 2957.01 (C-H stretching), 3418.20 (OH stretching) and 3783.79 (NH stretching). ^1^H NMR (d_6_-DMSO, 600 MHz): *δ*_H_ 0.76–0.79 (18 H, m, (CH_3_)_2_–Leu^2,9,12^), 0.79–0.98 (12H, m, *γ*-CH_3_–IIe^5^, *δ*-CH_3_–IIe^5^, (CH_3_)_2_–Val^6^), 1.24–1.62 (18H, m, CH_2_–Leu^2,9,12^, *γ*-CH–Leu^2,9,12^, *β*-CH –IIe^5^, *γ*-CH_2_–Met^8^, *β*-CH_2_–Lys^11^, *γ*-CH_2_–Lys^11^, *δ*-CH_2_–Lys^11^), 1.69 (1H, m, *γ*-CH–IIe^5^), 1.90 (1H, m, *β*-CH–Val^6^), 1.99 (3H, s, *δ*-CH_3_–Met^8^), 2.39 (2H, m, *β*-CH_2_–Met^8^), 2.71 (2H, m, Δ-CH_2_–Lys^11^), 2.77–3.10 (6H, m, CH_2_–Phe^1,13^*, β*-CH_2_–Tyr^10^), 3.55–3.61 (2H, m, CH_2_–Ser^3^), 3.67–3.82 (4H, m, CH_2_–Gly^4,7^), 4.04–4.25 (5H, m*, α*-CH–Phe^1^*, α*-CH–IIe^5^, *α*-CH–Val^6^, *α*-CH–Leu^9,12^), 4.26–4.46 (6H, m, Leu^2^, CH–Ser^3^, CH–Met^8^, *α*-CH–Tyr^10^, CH–Lys^11^, CH–Phe^13^), 5.05 (1H, t, OH–Ser^3^), 7.01–7.25 (14H, m, CH_Ar_–Phe^1,13^, CH_Ar_–Tyr^10^), 7.64–8.01 (5H, m, NH–Val^6^, NH–Met^8^, NH–Leu^9,12^, NH–Tyr^10^), 8.06–8.11 (4H, m, NH–Gly^4,7^, NH–IIe^5^, NH–Lys^11^), 8.17–8.60 (3H, m, NH–Leu^2^, NH–Ser^3^, NH–Phe^13^), 9.17, (1H, bs, OH–Tyr^10^). ^13^C-NMR (d_6_-DMSO, 150 MHz): δ ppm 10.9, 14.6, 15.3, 18.3, 19.2, 21.5, 21.6, 22.0, 22.8, 22.9, 23.1, 24.0, 24.2, 26.6, 27.1, 29.4, 30.2, 31.3, 31.9, 36.5, 37.1, 37.6, 37.8, 49.2, 51.0, 51.3, 51.4, 51.9, 52.0, 53.2, 53.4, 53.5, 54.5, 55.1, 56.6, 56.7, 58.1, 61.7, 61.8, 73.0, 114.7, 126.2, 127.1, 127.2, 127.7, 128.0, 128.5, 129.2, 129.5, 130.1, 134.7, 137.6, 155.8, 157.7, 157.9, 167.8, 168.2, 168.5, 168.9, 169.7, 170.2, 170.4, 170.5, 170.8, 171.0, 171.1, 171.2, 171.3, 171.4, 171.5, 171.6, 171.7, 171.8. LRMS (ESI) *m*/*z*: 1486.86/744.1 [M + 2H]^2+^.

### 4.3. Peptide Purification

To purify the peptides, RP-HPLC was carried out using an LC-908 W (Japan Analytical Industry, Tokyo, Japan) with a column of ODS-MAT80 (C18), which was eluted at a flow rate of 4 mL/min using a solution of 0.1%TFA in 40% H_2_O and 60% ACN. The purity of the peptides was determined using UPLC (Figure 1), and further characterisation was performed using various spectral techniques.

### 4.4. Circular Dichroism (CD) Analysis

The far-ultraviolet CD spectra were obtained using a JASCO J-810 spectropolarimeter (Jasco, Tokyo, Japan). In addition to this, the quartz cuvette of 10 mm path length was used. The temperature was maintained at 22 °C, and the output of the instrument was calibrated using D-(+)-10-camphorsulfonic acid. The peptides, which were present at a concentration of 1.0 mg/mL, were dissolved completely in 20 mM SDS at a concentration of 15 µM [21]. The instrument was set to record wavelengths ranging from 190 to 260 nm with a bandwidth of 2 nm. Ten consecutive scans (accumulations) were recorded for each spectrum at a rate of 50 nm/min, and the baseline was acquired under the same conditions. The secondary structures of all peptides were analysed based on the obtained CD spectra, as shown in Figure 2.

### 4.5. NMR Analysis

The nuclear magnetic resonance spectra were acquired with a “Bruker Avance NEO” apparatus. The frequency was set at 600 MHz. The temperature was maintained at 298 K, and the solvent used was DMSO, which was deuterated to avoid the moisture peaks of water. The peptides were present at a concentration of 4.0 mg to 5.0 mg. Sixteen scans were used, and the D1 time was kept at 2 s. Moreover, the data path (TD) was kept at 64 K.

### 4.6. Antimicrobial Assay

The peptides were assessed for their antibacterial activity by determining the minimum inhibitory concentration (MIC) using the two-fold serial dilution method following the National Committee of Clinical Laboratory Standards, as previously described [22]. Gram-negative (*Acinetobacter baumannii* (DSM 30007), *Escherichia coli* (ATCC 8739), *Enterobacter cloacae* (DSM 30054), *Helicobacter pylori* (ATCC 43504), *Klebsiella pneumoniae* (DSM 26371), *Pseudomonas aeruginosa* (ATCC 9027)) and Gram-positive bacteria (*Bacillus subtilis* (ATCC 6633), *Enterococcus faecalis* (DSM 2570), *Enterococcus faecium* (DSM 20477), *Staphylococcus aureus* (ATCC 6538)) were used for the antibacterial activity assays. In brief, 3 mL of Mueller–Hinton (MH) broth was inoculated with single clusters of the various bacterial strains and incubated overnight at 37 °C with agitation at 200 rpm. The next day, the bacterial solution was diluted with sterile MH by 1/100 and further incubated at 37 °C and 200 rpm until the bacteria reached the log phase growth (OD600nm around 0.6). The bacterial suspension was then adjusted to a density of approximately 10^5^ bacteria/mL, and 100 μL was added per well of 96-well microplates (Greiner BioOne, Les Ulis, France), followed by exposure to increasing concentrations of peptides (from 0 to 100 µM, 1:2 dilution). Plates were then incubated at 37 °C for 18-24 h under either aerobic (for *A. baumannii*, *E. cloacae*, *E. coli*, *P. aeruginosa*, *B. subtilis*, and *S. aureus*) or micro-aerobic conditions generated using GasPak units (for *E. faecalis, E. faecium* and, *H. pylori*). The MIC was determined as the lowest concentration of peptides that completely inhibited the visible growth of the bacteria. The optical density (OD) at 600nm was measured using a microplate reader (Biotek, Synergy Mx, Les Ulis, France) after the final incubation. All experiments were performed in triplicate (*n* = 3).

### 4.7. Antiproliferative and Cytotoxicity Assay

The antiproliferative activity of the peptides was evaluated, as previously described [23] using human cancer cell lines (A549 (lung), A2780 (ovary), HepG-2 (liver), MCF-7 (breast), MiaPaCa-2 (pancreas), MNT-1 (skin), and PC-3 (prostate)) and normal human lung fibroblasts (IMR-90) or epithelial cells (BEAS-2B). The cells were cultured in DMEM supplemented with 10% FBS, 1% L-glutamine, and 1% antibiotics at 37 °C in a 5% CO2 incubator. For the antiproliferative assay, cells were seeded in 96-well plates at approximately 3000 cells/well. Twenty-four hours later, cells were treated with increasing concentrations of peptides (1:2 serial dilution from 100 µM to 0 µM) for 48 h. After treatment, the number of viable cells was determined using a resazurin-based assay, and the fluorescence intensity was measured with a microplate reader (Ex 530 nm/Em 590 nm) [23]. The IC_50_ values of the peptides (i.e., the concentrations causing 50% inhibition of cell proliferation) were determined using GraphPad**^®^** Prism 7 software. The experiments were performed in triplicate (*n* = 3). The cytotoxicity of the peptides was also evaluated using confluent/no-dividing cells, as previously described [22]. Briefly, human cells were seeded onto 96-well plates at 20,000 cells/well. When they reached 80–90% confluence (after 24–48 h), cells were treated with escalating concentrations of peptides. After 48h incubation, cell viability was measured using a resazurin assay and CC_50_ (i.e., the concentrations causing 50% inhibition of cell viability) and determined using GraphPad**^®^** Prism 7 software. The experiments were performed in triplicate (*n* = 3).

## 5. Conclusions

Second-generation analogs of natural peptide—temporin-SHa (**1**)—were synthesised by substituting its glycine at position-10 with atypical amino acids like D-phenylalanine, D-naphthyl alanine and D-tyrosine. The new analogs had broad-spectrum antibacterial properties and featured potent activities against both Gram-negative (*Acinetobacter baumannii* DSM 30007, *Helicobacter pylori* ATCC 43504) and Gram-positive (*Bacillus subtilis* ATCC 6633, *Enterococcus faecalis* DSM 2570 and *Enterococcus faecium* DSM 20477) bacteria. Furthermore, these peptides demonstrated significant activity against various cancer cell lines, including lung cancer (A549), skin cancer (MNT-1), prostate cancer (PC-3), pancreatic cancer (MiaPaCa-2), and breast cancer (MCF-7) cells, with the IC_50_ value being in the range of 3.6–6.8 µM. Overall findings highlight the potential of second-generation temporin-SHa analogs as promising candidates to develop new broad-spectrum antibacterial and anticancer agents.

## Data Availability

All data are shown in the manuscript.

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
