# Peer review of "Synthesis of Second-Generation Analogs of Temporin-SHa Peptide Having Broad-Spectrum Antibacterial and Anticancer Effects"

_antibiotics, 2024, doi:10.3390/antibiotics13080758_

Round 1

Reviewer 1 Report

Comments and Suggestions for Authors

In general, the manuscript requires significant improvements. The English is poorly written and needs thorough revision. The primary motivation for the suggested modifications is to enhance the clarity, accuracy, and overall quality of the manuscript.

Results Section:

The results section is very confusing and needs significant clarification. Ensure that the results are presented logically and coherently, making it easy for readers to follow the study's findings and conclusions.

  1. The figure 1 was not mentioned in the text. It needs to be referenced and discussed within the results section. Ensure that all figures are referenced and discussed in the text.
  2. The wavelength range mentioned is 185-400 nm, whereas the materials and methods section specifies 185-260 nm. This discrepancy should be clarified.
  3. The manuscript does not explain how the secondary structure was determined. Details on the methodology used should be included.
  4. The secondary structure of the peptide in solution is not described. This information is crucial and should be added.
  5. The authors evaluated the peptide in the presence of SDS but did not explain the rationale. Additionally, the concentration of SDS used is not provided. Both the reason for using SDS and its concentration should be clearly stated.
  6. The manuscript refers to CD units as "gmed," which is unconventional. CD signals are typically expressed as ellipticity (mdeg) or mean residue ellipticity (deg/cm²/dmol). This should be corrected for clarity.
  7. The figure 2 is not mentioned in the text. It should be referenced and discussed.
  8. The figure 3 is confusing as it does not clearly differentiate between cancer cells and normal cells. The figure should be revised to separate and clearly label these cell types.

 Materials and Methods Section:

  • number of accumulations performed for CD spectra is not specified. This detail is necessary for reproducibility and should be included.

After these improvements, the manuscript should be thoroughly re-evaluated.

Comments on the Quality of English Language

The English is poorly written and needs thorough revision.

Author Response

Comments of Reviewer 1:

In general, the manuscript requires significant improvements. The English is poorly written and needs thorough revision. The primary motivation for the suggested modifications is to enhance the clarity, accuracy, and overall quality of the manuscript.

Dear Sir, Thank you for your valuable feedback on our manuscript. We have taken your comments into consideration and have worked to improve the English language and overall clarity of the paper. To ensure the highest standard, the manuscript has been thoroughly reviewed and edited by a scientist (PhD) and english native: Mr John Connolly (JohnConnolly20@rcsi.ie) who is collaborating with Dr Marc Maresca (from Department of Chemistry, RCSI University of Medicine and Health Sciences, Dublin, Ireland). We believe these revisions have significantly enhanced the readability and quality of our manuscript. We hope you will find the updated version satisfactory. Thank you for your time and consideration.

Result Section:

The results section is very confusing and needs significant clarification. Ensure that the results are presented logically and coherently, making it easy for readers to follow the study's findings and conclusions.

  1. The figure 1 was not mentioned in the text. It needs to be referenced and discussed within the results section. Ensure that all figures are referenced and discussed in the text.

Although figure 1 was mentioned in section 4.3, we have now mentioned it in result section (section 2.1) as per the suggestion of reviewer. Please see line no. 127–128 of the revised manuscript.

  1. The wavelength range mentioned is 185-400 nm, whereas the materials and methods section specifies 185-260 nm. This discrepancy should be clarified.

Thank you for the correction. We have rectified the method section which now specifies the wavelength range (185–400 nm). Please see line no. 518.

  1. The manuscript does not explain how the secondary structure was determined. Details on the methodology used should be included.

A paragraph was added in results sub-section 2.2 to explain how the secondary structure was determined. Please see line no. 140–150. Details on the methodology used are presented in Materials and Methods sub-section 4.4. Please see line no. 523–533.

  1. The secondary structure of the peptide in solution is not described. This information is crucial and should be added.

The secondary structure of the peptides in solution along with their percentages is now described in results sub-section 2.2 and in Table 3. Comments on the secondary structure and activity have been also added into the discussion section.

  1. The authors evaluated the peptide in the presence of SDS but did not explain the rationale. Additionally, the concentration of SDS used is not provided. Both the reason for using SDS and its concentration should be clearly stated.

SDS has been used for decade to mimic the membrane environment when conducting CD or NMR experiments.

The following text was also inserted in section 4.4 to show the concentration of SDS used:

“The peptides, which were present at a concentration of 1.0 mg/mL, were dissolved completely in 20 mM SDS at a concentration of 15 µM”. Please see line no. 515–517 and reference no 21.

  1. The manuscript refers to CD units as "gmed," which is unconventional. CD signals are typically expressed as ellipticity (mdeg) or mean residue ellipticity (deg/cm²/dmol). This should be corrected for clarity.

Corrected. Now CD units are expressed as “mdeg” throughout the manuscript. Thank you for pointing this out.

  1. The figure 2 is not mentioned in the text. It should be referenced and discussed.

Figure 2 has now been mentioned/referenced in section 2.2 and section 4.4. This figure has also been discussed in section 2.2

  1. The figure 3 is confusing as it does not clearly differentiate between cancer cells and normal cells. The figure should be revised to separate and clearly label these cell types.

Accordingly to reviewer’s suggestion, we created a new figure and cancer cells are shown in Fig 3 whereas normal cells are shown in Fig 4 in the revised version of the manuscript.

  1. Number of accumulations performed for CD spectra is not specified. This detail is necessary for reproducibility and should be included. After these improvements, the manuscript should be thoroughly re-evaluated.

Please see line no. 518–520 in section 4.4: Four consecutive scans (accumulations) were recorded for each spectrum at a rate of 100 nm/min, and the baseline was acquired under the same conditions.

Reviewer 2 Report

Comments and Suggestions for Authors

Dear authors

the work is a preliminary attempt to assess a sar study; one-by one changes in Xaa has been done to study the biological and functional changes in activity and structure of peptide.

i think it is interesting and deserves consideration in this location, but some issues are due: please better explain previus SAR studies if any; add overall yields for each product; please consider to add some reference on recent antimicrobial peptides developed via SPPS for example: "New Teixobactin Analogues with a Total Lactam Ring"

Author Response

Comments of Reviewer 2 :

Dear authors

The work is a preliminary attempt to assess a SAR study; one-by one changes in Xaa has been done to study the biological and functional changes in activity and structure of peptide. I think it is interesting and deserves consideration in this location, but some issues are due: please better explain previous SAR studies if any;

Dear Sir,

Thank you for your positive feedback on our work. We appreciate your recognition of the significance of our work and your support for its consideration in the journal.

Previous studies only reported anti-cancer activities of [G10a]-SHa analog which were described in comparison to 2nd generation analogs and mentioned in reference no 13, line no. 224. We have performed the anti-cancer and anti-microbial of these 2nd generation analogs for the first time.

Add overall yields for each product;

Thank you for pointing this out. We have now included overall yields of each product in Table 2.

Please consider to add some reference on recent antimicrobial peptides developed via SPPS for example: "New Teixobactin Analogues with a Total Lactam Ring"

We’ve now added some references on recent antimicrobial peptides developed via SPPS, including the reference for "New Teixobactin Analogues with a Total Lactam Ring". Please see line no. 81–91 reference no. [17], [18], [19] and [20].

Some Additional Changes

Table 1 has been shifted to page 3.

Following irrelevant paragraph was deleted from section 2.2

“This results in the insertion into anionic outer layer of bacterial cytoplasmic membrane, which ultimately causes the permeabilization or disruption via barrel-steve or carpet-like mechanism. Peptides generate strong interaction with anionic phosholipids or negatively charged lipopolysaccharides through Coulombic interactions”.

Changes / corrections done in the revised manuscript are highlighted in red colored text.

Round 2

Reviewer 1 Report

Comments and Suggestions for Authors

The manuscript has been revised and improved by the authors. However, the quality of the CD spectra could be further enhanced. The current signal-to-noise ratio is suboptimal due to insufficient accumulation and inappropriate scanning speed. To improve the data quality, it is recommended that the authors conduct the experiment with a higher number of accumulations (10) and a reduced scanning speed (50 nm/min). Additionally, applying data smoothing could further enhance the clarity of the spectra. "Another aspect to consider is the wavelength range used for data collection. It is recommended to clearly present the data within the 185-260 nm range, as this range provides essential information about the spectral features relevant to the study.

Author Response

Dear Reviewer,

We would like to thank you for your valuable comments and suggestions. Please find below our answers. 

Regards

Dr M Maresca

Response to Reviewer # 2 (Round 2)

The manuscript has been revised and improved by the authors. However, the quality of the CD spectra could be further enhanced. The current signal-to-noise ratio is suboptimal due to insufficient accumulation and inappropriate scanning speed. To improve the data quality, it is recommended that the authors conduct the experiment with a higher number of accumulations (10) and a reduced scanning speed (50 nm/min). Additionally, applying data smoothing could further enhance the clarity of the spectra. "Another aspect to consider is the wavelength range used for data collection. It is recommended to clearly present the data within the 185-260 nm range, as this range provides essential information about the spectral features relevant to the study.

Answer: Thank you very much for the encouraging comments. We have conducted the experiment with 10 accumulations and reduced the scanning speed to 50 nm/min. Data smoothing was applied to enhance the clarity. In our manuscript, the data is presented in the range of 190–260 nm instead of 185–260 nm due to instrument/software limitation. The figure 2 of the previous manuscript has also been replaced with a new figure 2. Results of the experiment are summarized by inserting a new table 3 on page no. 6. In this revised version of round 2, all the text additions / modifications in the manuscript are highlighted in yellow highlighted red colored text.